# Multibody Model with Foot-Deformation Approach for Estimating Ground Reaction Forces and Moments and Joint Torques during Level Walking through Optical Motion Capture without Optimization Techniques

**DOI:** 10.3390/s24092792

**Published:** 2024-04-27

**Authors:** Naoto Haraguchi, Kazunori Hase

**Affiliations:** Department of Mechanical Systems Engineering, Tokyo Metropolitan University, Tokyo 191-0065, Japan

**Keywords:** biomechanical analysis, motion analysis, inverse kinematics, inverse dynamics, musculoskeletal model, contact model

## Abstract

The biomechanical-model-based approach with a contact model offers advantages in estimating ground reaction forces (GRFs) and ground reaction moments (GRMs), as it does not rely on the need for training data and gait assumptions. However, this approach faces the challenge of long computational times due to the inclusion of optimization processes. To address this challenge, the present study developed a new optical motion capture (OMC)-based method to estimate GRFs, GRMs, and joint torques without prolonged computational times. The proposed approach performs the estimation process by distributing external forces, as determined by a multibody model, between the left and right feet based on foot deformations, thereby predicting the GRFs and GRMs without relying on optimization techniques. In this study, prediction accuracies during level walking were confirmed by comparing a general analysis using a force plate with the estimation results. The comparison revealed excellent or strong correlations between the prediction and the measurements for all GRFs, GRMs, and lower-limb-joint torques. The proposed method, which provides practical estimation with low computational cost, facilitates efficient biomechanical analysis and rapid feedback of analysis results, contributing to its increased applicability in clinical settings.

## 1. Introduction

In the fields of rehabilitation and sport, analyzing ground reaction forces (GRFs) and ground reaction moments (GRMs) offers advantages in assessing the risk of musculoskeletal disorders and evaluating physical performance [1,2]. To address the challenges associated with the complexity of recording GRFs and GRMs, previous studies have developed simplified measurement techniques that do not require a force plate. One such technique involves wearable systems for recording GRFs and GRMs, such as pressure insoles or instrumented force shoes [3]. These systems offer advantages in recording GRFs and GRMs with fewer limitations on measurement location. However, concerns remain regarding their low durability and the impact of the weight and height of the wearable instruments on the motion of participants.

Another technique for measuring GRFs involves a prediction approach using kinematic data recorded by optical motion capture (OMC) or inertial measurement units (IMUs) [3,4]. The OMC-based method contributes to high accuracy in GRF estimation because it predicts GRFs based on the body-segment positions obtained with high precision by OMC which correctly records three-dimensional positions. While the IMU-based method faces challenges in accurately estimating GRFs from kinematics data, as IMUs cannot directly record body-segment positions, they have the advantage of fewer limitations on measurement location. Researchers choose measurement systems based on the accuracy and applicability required for their estimation system, as well as the computational approach. Many GRF estimation techniques include two computational approaches: the statistical-model-based method or the biomechanical-model-based method. The statistical-model-based approach enables the prediction of GRFs and GRMs with high accuracy, relying on statistical algorithms such as machine learning [5,6,7,8,9,10]. Recently, a GRF estimation method combining electromyography measurements and machine learning has also been proposed [11], and the statistical-model-based approach is rapidly developing as an estimation technique. However, this method has a disadvantage in its application to situations where obtaining sufficient training data is challenging, thus limiting the range of movements that can be analyzed.

The biomechanical-model-based approach enables the estimation of GRFs and GRMs using multibody dynamics calculations [3,4]. Although this method has the potential to be applied in various movements, it faces limitations in making accurate predictions during the double-support phase due to the closed-loop structure between humans and the ground. To address this challenge, several studies have proposed computational techniques within the biomechanical-model-based approach. While artificial neural networks and smooth transition assumptions have been employed as accurate estimation methods [12,13], they lack applicability due to the need for training data and gait assumptions. In contrast, some studies have proposed computational techniques that incorporate a contact model in which the contact state between the foot and the ground is represented by a mathematical model, such as a spring-damper model [14,15]. Since this method calculates GRFs and GRMs from the measured contact state between the foot and the ground, it does not require any training data or gait assumptions, making it applicable to activities other than walking [16,17].

Previously, we developed the contact-model-based approach for predicting GRFs, GRMs, and joint torques [18]. This method employs multibody dynamics calculations based on kinematic data for GRF prediction without requiring training data or making gait assumptions, thus providing advantages in terms of versatility and applicability. However, our study faced a limitation of significant computational cost due to the optimization technique to tune the parameters necessary for estimating GRFs and GRMs. Similarly, other studies using the contact-model-based approach also rely on optimization processes to tune parameters such as the coefficients of the spring-damper model [14,15,16,17]. While alternative methods exist for estimating GRFs and GRMs using different computational techniques [19,20], these methods also incorporate optimization techniques for GRF prediction. Therefore, a prediction method focused on reducing computational time could significantly contribute to the clinical applications of GRF estimation, such as improving the efficiency of the biomechanical analysis and rapidly presenting analysis results to medical staff and patients.

This study developed a new prediction method for GRFs, GRMs, and joint torques with low computational cost by focusing on deformations of the foot alignment. Previous studies have reported that the foot arch and soft tissues contribute to foot deformations under load, reaching magnitudes of millimeters or centimeters under the load of the human body mass [21,22]. Based on this fact, we hypothesized that foot deformations can be recorded with OMC, which correctly obtains three-dimensional positions, and that the load condition of the foot can be predicted from the recorded foot deformations. Furthermore, if the load condition can be estimated from foot deformations, GRFs and GRMs during the double support phase can be predicted without optimization techniques by distributing external forces calculated by the biomechanical model according to the load condition. Therefore, this study developed a new OMC-based prediction method for GRFs, GRMs, and joint torques using a hybrid approach of a multibody-dynamics model and foot deformations and evaluated the estimation accuracy and its advantages for biomechanical analysis.

## 2. Materials and Methods

### 2.1. Participants

The study participants were 18 volunteers (10 males, 8 females; mean age: 23.3 ± 2.5 years; mean height: 1.67 ± 0.10 m; mean weight: 55.2 ± 7.46 kg) with no history of musculoskeletal disorders. Approval for the study was obtained from the Ethics Committee of Tokyo Metropolitan University. Prior to the experiment, all participants received both verbal and written explanations regarding the study’s content, and they provided written informed consent.

### 2.2. Conditions

As the initial step in developing the prediction method, this study evaluated the accuracy of estimation during level walking with bare feet, a fundamental human motion. The experiment included three walking speeds: normal, fast, and slow. The normal speed was adjusted to each participant’s preferred walking pace, while the fast and slow speeds were approximately +20% and −20% of the normal walking speed, respectively. Before the experiment, participants were familiarized with each walking speed.

### 2.3. Measurements

The three-dimensional coordinates, denoted as *x* for the anterior−posterior axis, *y* for the vertical axis, and *z* for the medio−lateral axis, of markers attached to the whole body were measured using an OMC system (OptiTrack Flex3; Natural Point Inc., Corvallis, OR, USA). A total of 49 reflective markers were placed on various body locations based on the marker set of the Gait 2392 musculoskeletal model in OpenSim [23,24,25,26]. The marker data were digitally filtered (low-pass filter, Butterworth fourth-order type, −3 dB at 6 Hz) and were sampled at 100 Hz.

To validate the OMC results, the GRF and GRM were measured using a force plate (TF-4060-D; Tec Gihan Co. Ltd., Kyoto, Japan). The force data were digitally filtered (low-pass filter, Butterworth fourth-order type, −3 dB at 18 Hz) and were sampled at 1000 Hz.

### 2.4. Data Analysis

#### 2.4.1. Kinematics

At the beginning of the prediction procedure, the kinematics of each joint, segment, and contact point were computed based on the three-dimensional coordinates of markers recorded by the OMC system. The joint and segment kinematics were calculated using the Gait 2392 musculoskeletal model in OpenSim [23,24,25,26]. This model consists of 12 segments, including the torso, pelvis, femurs, tibias, taluses, calcanei, and toes, and had 23 degrees of freedom (DOFs). The segment reference frame and joint coordinate system were defined based on previous studies in the lower extremity [23], plantar knee [24], and low back model [25]. The hip and back joints had three DOFs expressed in Euler angles with the sequence *Z* (flexion−extension) *X* (abduction−adduction) *Y* (rotation). The knee, ankle, subtalar, and metatarsophalangeal joints each had one DOF in flexion−extension or dorsiflexion−plantarflexion. The pelvis had six DOFs for global coordinates, which included three translational DOFs and three rotational DOFs expressed in Euler angles with the sequence *Z* (flexion−extension) *X* (abduction−adduction) *Y* (rotation). The inverse kinematics using the model in OpenSim computed the joint kinematics vector $q\in R^{23}$, which includes the joint angles of all joints, and the segment kinematics vectors $r_{x}\in R^{12}$, $r_{y}\in R^{12}$, and $r_{z}\in R^{12}$, which include the translational position of the center of mass (COM) of all segments [27].

To compute GRFs and GRMs based on foot deformations, a total of 20 contact points (10 per foot) were positioned on the calcaneus and toe segments, as shown in Figure 1. The segmentations of the calcaneus and toe was defined based on the Gait 2392 musculoskeletal model in OpenSim [23]. The horizontal locations of the contact points were derived from the marker positions based on the marker set of the Gait 2392 musculoskeletal model in OpenSim. The vertical locations of the contact points were set to have an offset of 30 mm toward the ground during static standing, using the OMC data of the static standing calibration with weight on the heel side. Subsequently, the contact point computation yielded translational positions of the contact points on the foot, represented by $p_{x}\in R^{20}$, $p_{y}\in R^{20}$, and $p_{z}\in R^{20}$.

#### 2.4.2. Ground Reaction Forces and Moments

Before computing GRFs and GRMs, the total external forces acting on the human body were calculated using the translational equations of motion. The external forces $F_{\mathrm{ext}\_x}$, $F_{\mathrm{ext}\_y}$, and $F_{\mathrm{ext}\_z}$ were expressed as follows:(1)$$F_{\mathrm{ext}\_x}=\sum_{s=1}^{12}m^{s}{\ddot{r}}_{x}^{s},$$
(2)$$F_{\mathrm{ext}\_y}=\sum_{s=1}^{12}m^{s}\left({\ddot{r}}_{y}^{s}-g\right),$$
(3)$$F_{\mathrm{ext}\_z}=\sum_{s=1}^{12}m^{s}{\ddot{r}}_{z}^{s},$$
where $r_{x}^{s}$, $r_{y}^{s}$, and $r_{z}^{s}$ represent the anterior, vertical, and medial position of the COM of the *s*th segment, which are components of $r_{x}$, $r_{y}$, and $r_{z}$, respectively; $m^{s}$ is the mass of the *s*th segment as defined in the previous study [25]; and g is gravitational acceleration.

Because the external forces, calculated from the equation of motion, represent the total forces applied to the human body, it is necessary to allocate these forces to each foot to calculate the GRFs and GRMs. The present study determined the amount of sinkage to the ground at the *i*th contact-point position $h^{i}$ based on the contact-point positions, and the total external force was distributed to each contact point based on the ratio of $h^{i}$ to the overall sinkage. The vertical GRF $F_{\mathrm{GR}\_y}$, frontal GRM $M_{\mathrm{GR}\_x}$, and sagittal GRM $M_{\mathrm{GR}\_z}$ applied to one foot were computed as follows:(4)$$h^{i}=\left\{\;\begin{array}{l} p_{y}^{i}-p_{\mathrm{crit}}\;\;\;\;\;\mathrm{if}\;p_{y}^{i}<p_{\mathrm{crit}}\;\mathrm{and}\;{\dot{p}}_{y}^{i}<v_{\mathrm{crit}}\\ 0\;\;\;\;\;\;\;\;\;\;\;\;\;\;\;\;\;\;\mathrm{otherwise} \end{array}\right.,$$
(5)$$F_{\mathrm{GR}\_y}=\sum_{i=1}^{10}\frac{h^{i}}{\sum_{k=1}^{20}h^{k}}F_{\mathrm{ext}\_y},$$
(6)$$M_{\mathrm{GR}\_x}=-\sum_{i=1}^{10}\frac{h^{i}}{\sum_{k=1}^{20}h^{k}}p_{z}^{i}F_{\mathrm{ext}\_y},$$
(7)$$M_{\mathrm{GR}\_z}=\sum_{i=1}^{10}\frac{h^{i}}{\sum_{k=1}^{20}h^{k}}p_{x}^{i}F_{\mathrm{ext}\_y},$$
where $p_{x}^{i}$, $p_{y}^{i}$, and $p_{z}^{i}$ represent the anterior, vertical, and medial position of the *i*th contact point, which are components of $p_{x}$, $p_{y}$, and $p_{z}$, respectively, and $p_{\mathrm{crit}}$ and $v_{\mathrm{crit}}$ are the critical position and velocity of the contact point, respectively, which were set to $p_{\mathrm{crit}}$ = 0 and $v_{\mathrm{crit}}$ = 0.05, as determined empirically. In this study, the ground height was set to $p_{y}^{i}$ = 0. The origin of the horizontal directions $p_{x}^{i}$ and $p_{z}^{i}$ was positioned at the ankle joint, as measured by OMC (the midpoint of the markers of the lateral and medial malleolus).

During the single-support phase, the anterior GRF $F_{\mathrm{GR}\_x}$ and medial GRF $F_{\mathrm{GR}\_z}$ applied to one foot were determined using the horizontal external forces calculated from the equation of motion, as follows:(8)$$F_{\mathrm{GR}\_x}=F_{\mathrm{ext}\_x},$$
(9)$$F_{\mathrm{GR}\_z}=F_{\mathrm{ext}\_z}.$$

During the double support phase, the anterior and medial GRFs were calculated using a different method compared to the single-support phase. Previous studies have reported that the GRF vector intersects a point located above the COM of the system, referred to as the virtual pivot point (VPP), in a dynamic system such as a human in motion [28]. Accordingly, the present study computed the horizontal GRF based on the assumption that the GRF vector intersects the VPP of the human body. First, three-dimensional positions of the center of pressure (COP) of the GRF, which is the starting point of the GRF vector, are calculated by the vertical GRF and the GRM around the horizontal axis, as follows:(10)$$p_{\mathrm{COP}\_x}=M_{\mathrm{GR}\_z}/F_{\mathrm{GR}\_y},$$
(11)$$p_{\mathrm{COP}\_y}=0,$$
(12)$$p_{\mathrm{COP}\_z}=-M_{\mathrm{GR}\_x}/F_{\mathrm{GR}\_y},$$
where the vertical position of the COP was set to the ground height, that is, $p_{\mathrm{COP}\_y}$ = 0. Then, if the GRF vector is assumed to intersect the VPP, the GRF can be expressed as a vector connecting the COP and the VPP multiplied by a real number. Consequently, the anterior GRF $F_{\mathrm{GR}\_x}$ and medial GRF $F_{\mathrm{GR}\_z}$ during the double support phase are calculated using the real number α, as follows:(13)$$\alpha =F_{\mathrm{GR}\_y}/\left(r_{\mathrm{VPP}\_y}-p_{\mathrm{COP}\_y}\right),$$
(14)$$F_{\mathrm{GR}\_x}=\alpha \left(r_{\mathrm{VPP}\_x}-p_{\mathrm{COP}\_x}\right),$$
(15)$$F_{\mathrm{GR}\_z}=\alpha \left(r_{\mathrm{VPP}\_z}-p_{\mathrm{COP}\_z}\right),$$
where $r_{\mathrm{VPP}\_x}$, $r_{\mathrm{VPP}\_y}$, and $r_{\mathrm{VPP}\_z}$ represent the anterior, vertical, and medial positions of the VPP of the entire body, respectively. The VPP position was defined as 37.5 mm upward along the axis direction represented in the trunk segment coordinate system from the COM of the whole body, after a previous experimental study [28]. The whole-body COM was determined from the segment kinematics vectors $r_{x}$, $r_{y}$, and $r_{z}$. Subsequently, the transverse GRM $M_{\mathrm{GR}\_y}$ was computed using the horizontal GRFs and the COP of the GRF, as follows:(16)$$M_{\mathrm{GR}\_y}=p_{\mathrm{COP}\_z}F_{\mathrm{GR}\_x}-p_{\mathrm{COP}\_x}F_{\mathrm{GR}\_z}.$$

The computational flow of GRFs and GRMs is summarized as follows. First, the positions of the COM of each segment and the contact points on the foot segment are derived from the OMC data. Based on these COM kinematics, the equation of motion determines the external forces acting on the human body. Through the foot deformation approach, these external forces are distributed by the positional relationship between the contact point and the ground, obtaining the vertical GRF, sagittal GRM, and frontal GRM. Following this computation, the anterior GRF and medial GRF, and COPs are calculated using different approaches for the single-support phase and double support phase. Finally, the transverse GRM is derived from the computed GRFs and COPs.

#### 2.4.3. Joint Torques

Joint torques were determined through inverse dynamics analysis using the estimated GRFs and GRMs. The joint torque vector $\tau \in R^{23}$ was computed as follows:(17)$$\tau =I\ddot{q}+\Gamma \left(q,\;\dot{q},F_{\mathrm{GR}\_x},F_{\mathrm{GR}\_y},F_{\mathrm{GR}\_z},M_{\mathrm{GR}\_x},M_{\mathrm{GR}\_y},M_{\mathrm{GR}\_z}\right),$$
where $I\in R^{23\times 23}$ represents the inertia matrix as defined in the previous study [25], and $\Gamma \in R^{23}$ is a vector consisting of Coriolis, centrifugal, gravitational, and external forces. The inverse dynamics analysis was conducted with the Gait 2392 musculoskeletal model in OpenSim [27], which is the same model used in the inverse kinematics analysis.

### 2.5. Accuracy and Sensitivity Analysis

To evaluate the prediction accuracy, the predicted GRFs and GRMs were compared with the force-plate data. The predicted joint torques were compared with an inverse dynamics solution using the Gait 2392 musculoskeletal model in OpenSim with the input measurement data [27]. The agreement between the prediction and measurement was evaluated using Pearson’s correlation coefficient (ρ), which was classified as follows: ρ≤0.35 for weak, 0.35<ρ≤0.67 for moderate, 0.67<ρ≤0.9 for strong, and 0.9<ρ for excellent correlation [29]. In addition, we computed the root-mean-square error (RMSE) and relative RMSE (rRMSE), which normalized the RMSE by the average peak-to-peak amplitude for the two solutions [13]. All statistical analyses were performed using MATLAB 9.9 (MathWorks, Inc., Natick, MA, USA).

In the proposed method, which relies on foot deformation, the measurement error of the contact point may have a significant impact on the estimation accuracy. Therefore, a sensitivity analysis was conducted to investigate the influence of the measurement error of the contact point on prediction accuracy. To simulate data with measurement errors, white noise was added to all contact-point positions, $p_{x}$, $p_{y}$, and $p_{z},$ using the *rand* function in MATLAB. This study compared the RMSE and rRMSE of the GRFs and GRMs under three conditions with maximum white noise amplitudes of 1, 10, and 100 mm.

## 3. Results

The GRF and GRM curves throughout a gait cycle during normal-speed walking are shown in Figure 2. At all three experimental walking speeds, excellent or strong correlations were observed for all GRFs and GRMs. A comparison of the estimation accuracies in GRFs and GRMs with those of previous studies is presented in Table 1.

The joint torque curves throughout a gait cycle during normal-speed walking are shown in Figure 3. At all three experimental walking speeds, excellent or strong correlations were observed for all joint torques. A comparison of estimation accuracies in joint torques with those of previous studies is presented in Table 2.

The RMSEs and rRMSEs of the GRFs and GRMs with white noise in contact-point position data during normal-speed walking are presented in Table 3. The sensitivity analysis confirms that the estimation accuracies of GRFs and GRMs tended to deteriorate as the amplitude of the white noise increased.

The computational time from OMC data input to GRFs and GRMs output was approximately 4 s for one trial (CPU: Intel Core i7-10700 with 4.0 GHz of average speed; memory: 32 GB; software: MATLAB 9.9 and OpenSim 4.1; OS: Windows 11 Home).

The results of the statistical analysis for each speed are summarized in Appendix A. The time-series data for each speed are included in Appendix A.

## 4. Discussion

The proposed method demonstrated estimation accuracy comparable to that of previous studies, as shown in Table 1 and Table 2 [12,13,16]. Thus, the proposed approach, which does not rely on optimization techniques, succeeded in achieving these estimation accuracies for GRFs, GRMs, and joint torques at a low computational cost of a few seconds. In previous studies that employed optimization techniques for prediction, the computational time was relatively long because the parameters for computing the estimated values had to be determined through exploratory calculations [18]. Although the computational time depends on various factors, it is assumed that optimization approaches requiring exploratory calculations involve a computational cost at least in the order of several minutes. Therefore, the proposed method offers advantages over conventional methods, such as rapid feedback of the analysis results, thereby contributing to the clinical application of these kinetic factor estimation methods.

Another feature of the proposed method is its potential applicability as a GRF estimation technique for various activities. In previous studies, the smooth transition assumption approach has been applied to GRF estimation only during walking [13,30]. Although the artificial neural network approach can be extended to GRF estimation for activities other than walking [31], its applicability might be constrained in situations where sufficient training data cannot be obtained. In contrast, because the present method operates independently of statistical models or walking assumptions, it has the potential to be applied to GRF estimation across a wide range of activities, such as sports, daily activities, and orthopedic or prosthetic conditions.

Here, we provide a detailed description of the prediction accuracy of the proposed method. Although previous studies estimated GRFs during the single-support phase using an approach relying on the equation of motion as used in our study, GRFs during the double support phase were calculated using complex computational techniques, such as a statistical model, walking assumptions, and optimization [12,13,14,15,16,17,18,19,20]. In contrast, our proposed estimation method calculated GRFs using only the external force derived from the equation of motion and the amount of sinkage at the contact point. Consequently, the prediction results of this study were more susceptible to the influence of measurement errors compared to conventional approaches. Hence, the simplicity of the computational method employed in this study had a detrimental impact by producing a lower prediction accuracy for the GRFs.

Previous studies estimated the GRMs using the equation of motion and some computational techniques, similar to their GRF prediction [12,13,14,15,16,17,18,19,20]. However, the estimated GRMs based on the rotational equation of motion could include modeling errors related to the mass, moment of inertia, and COM of each body segment. These errors potentially degrade the estimation accuracy compared to the GRF prediction based on the translational equation of motion, which only incorporates modeling errors of the mass of each body segment. On the contrary, the proposed estimation method computed the GRMs using the estimated GRFs and the position of the contact point, thereby eliminating the modeling errors associated with the moment of inertia and COM of each body segment from the GRM prediction. Consequently, the accuracy of GRM estimation for the entire gait cycle was improved compared to previous studies.

The proposed method provided estimation accuracies in joint torques comparable to those of the estimation method based on the smooth transition assumption [13]. In addition, a previous study has reported that the optimization approach provides joint torque prediction accuracy levels similar to those of the smooth transition assumption [16]. Therefore, while the proposed method does not achieve an estimation accuracy comparable to the artificial neural network approach, which provides superior estimation accuracy compared to other methods, the present study demonstrates a practical accuracy of the proposed approach comparable to that of the smooth transition assumption and the optimization approach.

While the GRF prediction was not significantly affected by white noise up to 10 mm, the GRM prediction showed a deterioration in accuracy at 10 mm of white noise. In the proposed prediction method, noise in the contact-point positions influenced the distribution of the external force computed by the equation of motion, but not for the magnitude of the external force. Therefore, the noise in the contact-point positions had less impact on the GRFs, which are calculated as the sum of the forces acting on the contact points of each foot. In contrast, the distribution of the external force at each contact point had a significant effect on the COP, leading to a deterioration in estimation accuracy for the GRMs. Based on the results of the sensitivity analysis, it is expected that the estimation accuracy of GRMs will decline if the proposed method is applied to IMU-based prediction, which has fewer limitations for measurement locations and lower measurement accuracy than that of the OMC-based approach. Nevertheless, the present method has the potential for application in IMU-based methods with practical accuracy because the estimation accuracy with white noise at 10 mm was not greatly different from the estimation accuracy in some previous studies, as shown in Table 3 [13,16].

Several limitations of this study are noted. Although the present study developed the GRF estimation method to address the location limitations of force plates, the proposed method can only be applied in situations where an OMC system can be installed. While the location limitations of the present method can potentially be relieved by employing IMUs instead of an OMC system, a critical challenge remains in determining the position of the foot contact points when using IMUs, as IMUs cannot directly measure three-dimensional coordinates.

In the present study, the experiment was conducted with participants in bare feet to accurately capture foot deformations. Consequently, this study cannot ensure prediction accuracy under conditions other than bare feet, such as when shoes are worn. To extend the applicability of the method to various motions in the future, the accuracy of this method should be confirmed under conditions other than bare feet.

This study estimated GRFs and GRMs for movement on level ground by assuming the critical position $p_{\mathrm{crit}}$ shown in Equation (4) to be constant. While the prediction accuracy needs confirmation, the proposed method may be applicable beyond level ground to situations where the critical position $p_{\mathrm{crit}}$ can be formulated, such as when the ground has a constant slope. However, it is challenging to apply this method to a rough walking surface with irregularities in the ground.

The estimation accuracy in this study was confirmed only for healthy young adults walking at three different speeds. Although the proposed method is expected to be applicable to various activities other than walking, further accuracy verification is necessary to apply the proposed method to those activities.

## 5. Conclusions

The proposed OMC-based method successfully estimated GRFs, GRMs, and joint torques with practical estimation accuracy while maintaining low computational costs. This achievement can be attributed to the reduction in biomechanical modeling errors in GRF prediction, which was realized through the foot-deformation approach. The proposed method, characterized by its low computational time without optimization techniques, is expected to enable clinical applications of GRF estimation. Furthermore, the proposed method has the potential for application to GRF estimation for various activities, as it operates independently of walking assumptions or training data. It is also applicable to IMU-based approaches, as it performs predictions under the influence of 10 mm of white noise.

## Figures and Tables

**Figure 1 sensors-24-02792-f001:**
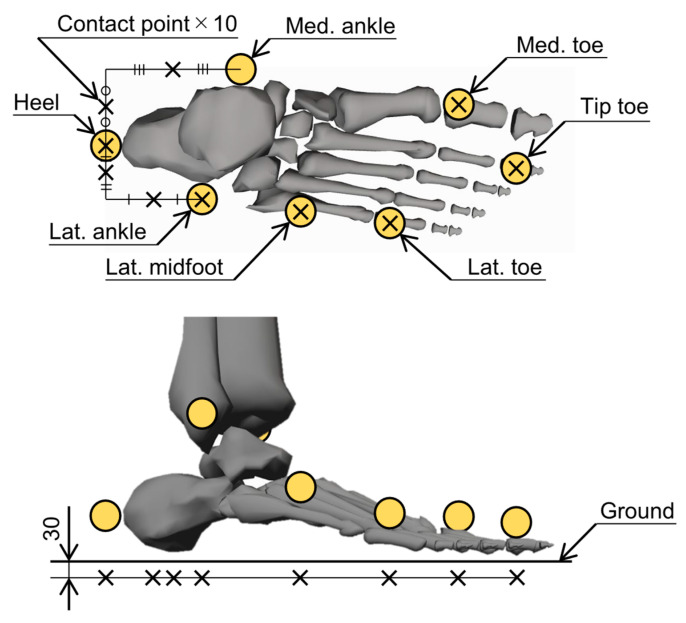
Locations of the contact points. The cross marks represent the contact points, and the circle symbols represent the reflective markers used in the optical motion capture. A total of 20 contact points, 10 per foot, were placed in the calcaneus and toe segments. The horizontal locations of the contact points were derived from the marker positions based on the marker set of the musculoskeletal model employed in this study, and the vertical locations of the contact points were set to have an offset of 30 mm toward the ground during static standing. Unit: mm.

**Figure 2 sensors-24-02792-f002:**
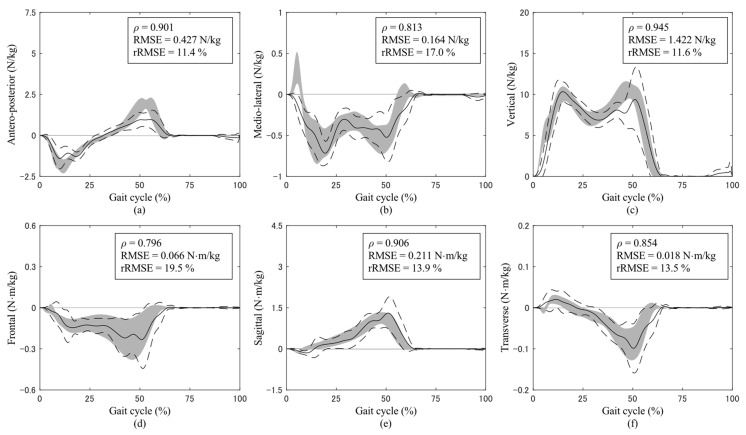
Ground reaction forces (GRFs) in the (**a**) antero−posterior axis with the anterior direction as positive, (**b**) medio−lateral axis with the lateral direction as positive, and (**c**) vertical axis with the upward direction as positive, and ground reaction moments (GRMs) in the (**d**) frontal plane around the anterior axis, (**e**) sagittal plane around the lateral axis, and (**f**) transverse plane around the vertical upward axis during normal-speed walking. The solid and dashed curves represent the average and standard deviation of the predictions, respectively, while the gray shading represents the average and standard deviation of the measurements. The upper right text of each graph shows Pearson’s correlation coefficient (ρ), the root-mean-square error (RMSE), and the relative RMSE (rRMSE). The magnitudes of the curves are normalized to the body mass of participants.

**Figure 3 sensors-24-02792-f003:**
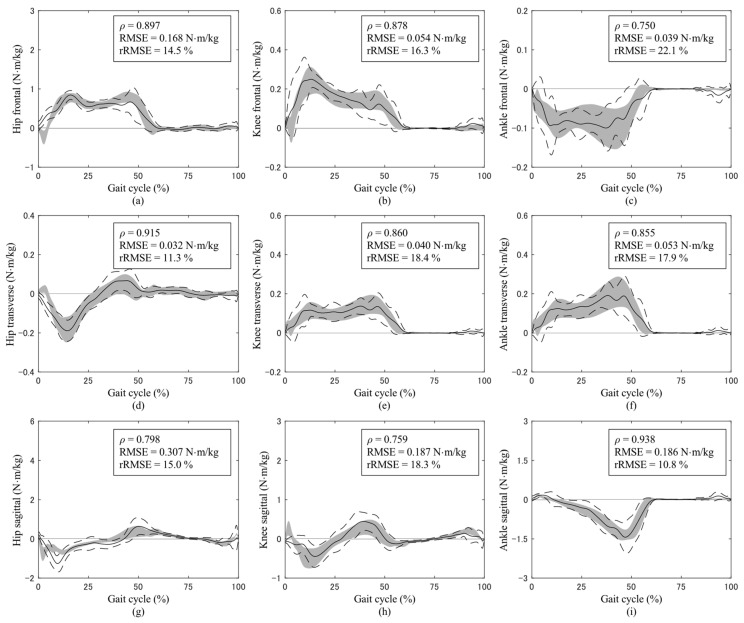
Joint torques in the frontal plane at the (**a**) hip, (**b**) knee, and (**c**) ankle with abduction/eversion torque as positive; in the transverse plane at the (**d**) hip, (**e**) knee, and (**f**) ankle with external rotation torque as positive; and in the sagittal plane at the (**g**) hip, (**h**) knee, and (**i**) ankle with flexion/dorsiflexion torque as positive. All joint torques are expressed in the proximal coordinate system during normal-speed walking. The solid and dashed curves represent the average and standard deviation of the predictions, respectively, while the gray shading represents the average and standard deviation of the measurements. The upper right text of each graph shows Pearson’s correlation coefficient (ρ), the root-mean-square error (RMSE), and the relative RMSE (rRMSE). The magnitudes of the curves are normalized to the body mass of participants.

**Table 1 sensors-24-02792-t001:** Comparison of root-mean-square errors (RMSEs) and relative RMSEs (rRMSEs) in ground reaction forces (GRFs) and moments (GRMs) during normal-speed walking with results of previous studies [12,13,16]. The average and standard deviation (SD) for all participants *N* are displayed, with magnitudes normalized to the body mass of participants.

Method	Artificial Neural Network [12]		Smooth Transition Assumption [13]		Optimization Approach [16]		Foot Deformation Approach (This Study)	
Participants	*N* = 5		*N* = 3		*N* = 9		*N* = 18	
	RMSE (SD) (N/kg or N·m/kg)	rRMSE (SD) (%)	RMSE (SD) (N/kg or N·m/kg)	rRMSE (SD) (%)	RMSE (SD) (N/kg or N·m/kg)	rRMSE (SD) (%)	RMSE (SD) (N/kg or N·m/kg)	rRMSE (SD) (%)
Anterior GRF	0.154 (0.057)	7.3 (0.8)	0.473 (0.068)	10.9 (0.8)	0.38 (0.07)	9.3 (2.0)	0.427 (0.135)	11.4 (3.9)
Medial GRF	0.040 (0.022)	10.9 (1.8)	0.191 (0.034)	20.0 (2.7)	0.17 (0.04)	14.9 (3.4)	0.164 (0.029)	17.0 (3.6)
Vertical GRF	0.649 (0.182)	5.8 (1.0)	0.710 (0.190)	5.6 (1.5)	0.74 (0.13)	6.6 (1.1)	1.422 (0.678)	11.6 (5.4)
Frontal GRM	0.052 (0.029)	22.8 (4.9)	0.148 (0.013)	32.5 (4.3)	0.11 (0.02)	22.9 (5.9)	0.066 (0.028)	19.5 (7.1)
Sagittal GRM	0.081 (0.045)	9.9 (1.9)	0.199 (0.106)	12.2 (4.8)	0.18 (0.05)	12.4 (3.5)	0.211 (0.080)	13.9 (7.9)
Transverse GRM	0.032 (0.018)	25.5 (4.5)	0.039 (0.015)	26.2 (9.4)	0.22 (0.06)	40.6 (11.3)	0.018 (0.007)	13.5 (5.4)

**Table 2 sensors-24-02792-t002:** Comparison of root-mean-square errors (RMSEs) and relative RMSEs (rRMSEs) in joint torques during normal-speed walking with results of previous studies [12,13]. The average and standard deviation (SD) for all participants *N* are displayed, with magnitudes normalized to the body mass of participants.

Method	Artificial Neural Network [12]		Smooth Transition Assumption [13]		Foot Deformation Approach (This Study)	
Participants	*N* = 5		*N* = 3		*N* = 18	
	RMSE (SD) (N·m/kg)	rRMSE (SD) (%)	RMSE (SD) (N·m/kg)	rRMSE (SD) (%)	RMSE (SD) (N·m/kg)	rRMSE (SD) (%)
Hip						
Frontal	0.052 (0.006)	5.1 (0.9)	0.106 (0.008)	9.9 (0.9)	0.168 (0.046)	14.5 (3.4)
Transverse	0.029 (0.040)	12.0 (1.0)	0.051 (0.006)	15.0 (1.2)	0.032 (0.007)	11.3 (2.6)
Sagittal	0.056 (0.041)	9.7 (2.0)	0.469 (0.067)	20.9 (2.1)	0.307 (0.095)	15.0 (2.5)
Knee						
Frontal	0.033 (0.019)	6.4 (1.6)	0.100 (0.017)	15.3 (2.8)	0.054 (0.023)	16.3 (6.4)
Transverse	0.043 (0.036)	13.8 (2.7)	0.042 (0.012)	25.4 (5.1)	0.040 (0.016)	18.4 (7.0)
Sagittal	0.020 (0.007)	8.1 (1.8)	0.307 (0.056)	18.7 (2.9)	0.187 (0.053)	18.3 (3.7)
Ankle						
Frontal	0.053 (0.028)	22.7 (5.0)	0.134 (0.012)	35.8 (4.6)	0.039 (0.016)	22.1 (7.2)
Transverse	0.033 (0.022)	25.0 (4.4)	0.039 (0.015)	26.1 (9.3)	0.053 (0.024)	17.9 (6.9)
Sagittal	0.091 (0.052)	10.5 (4.8)	0.190 (0.112)	9.7 (4.8)	0.186 (0.096)	10.8 (8.5)

**Table 3 sensors-24-02792-t003:** Root-mean-square errors (RMSEs) and relative RMSEs (rRMSEs) of ground reaction forces (GRFs) and moments (GRMs) with three levels of white noise in the contact-point position data during normal-speed walking. The average and standard deviation (SD) for all participants are displayed, with magnitudes normalized to the body mass of participants.

Noise Level	±1 mm		±10 mm		±100 mm	
	RMSE (SD) (N/kg or N·m/kg)	rRMSE (SD) (%)	RMSE (SD) (N/kg or N·m/kg)	rRMSE (SD) (%)	RMSE (SD) (N/kg or N·m/kg)	rRMSE (SD) (%)
Anterior GRF	0.426 (0.134)	11.4 (3.9)	0.432 (0.138)	11.6 (4.1)	0.739 (0.098)	24.1 (2.1)
Medial GRF	0.165 (0.029)	17.1 (3.6)	0.178 (0.029)	18.3 (4.3)	0.213 (0.045)	22.7 (3.5)
Vertical GRF	1.414 (0.678)	11.5 (5.4)	1.384 (0.680)	11.2 (5.4)	3.587 (0.397)	31.6 (4.1)
Frontal GRM	0.068 (0.029)	19.7 (7.4)	0.105 (0.036)	27.3 (10.8)	0.397 (0.085)	46.6 (9.3)
Sagittal GRM	0.213 (0.080)	14.0 (7.8)	0.245 (0.075)	15.6 (7.2)	0.420 (0.105)	25.8 (6.2)
Transverse GRM	0.019 (0.007)	13.7 (5.5)	0.026 (0.009)	16.9 (6.3)	0.036 (0.009)	19.5 (4.0)

## Data Availability

The data presented in this study are available as Appendix A.

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
