# Peer review of "Multibody Model with Foot-Deformation Approach for Estimating Ground Reaction Forces and Moments and Joint Torques during Level Walking through Optical Motion Capture without Optimization Techniques"

_sensors, 2024, doi:10.3390/s24092792_

Round 1
Reviewer 1 Report
Comments and Suggestions for Authors
This paper proposed a method to estimate GRF and moments and joint Torques of human walking based on multibody model. The topic is interesting. The paper is well organized. I scrutinized the formulation and validation, to my happiness, no mistake was found. The paper can be accepted for publication once the following concerns could be well addressed:
1. GRF is always relates to the motion of human body. Why the authors relate it to the foot deformation? What are the advantages by do this?
2. The authors claimed that the limitations of previous study is the computational cost due to the optimization technique to tune the parameters necessary for estimating GRFs and GRMs. Based on the multibody model, one can easily obtain the GRF once the motion of each segment can be determined.
3. RMSE presented in Table 2 is high. Try to explain the possible reasons.
Reviewer 2 Report
Comments and Suggestions for Authors
The study by Haraguchi and Hase focuses on developing a physically based estimation model for ground reaction forces during gait. Overall, the quality of the paper is very good, and the authors adeptly frame the literature, emphasizing the importance of using biomechanics approaches to obtain reliable information about how humans perform essential motor tasks, such as walking.
However, I have a few concerns related to the study. Specifically, these comments pertain to reconciling the ISB standards for force and joint moment description/visualization and clarifying the statistical tests used. In the following section, I outline both major and minor concerns for the authors’ consideration.
Major Concerns
1) In kinematics analysis, particularly in the methods section, it is crucial to emphasize and verify whether the definition of the segment reference frame and the parametrization of joint angles align with the ISB standards proposed by Wu, Ge, et al. in their article titled ‘ISB recommendation on definitions of joint coordinate system of various joints for the reporting of human joint motion—part I: ankle, hip, and spine’ (Journal of Biomechanics, 35.4, 2002, pp. 543-548). Furthermore, accurate definitions of joint angles and reference frames play a functional role in projecting forces (and subsequently plotting force vectors) relative to proximal or distal reference frames. While these aspects are sometimes left to the readers’ interpretation, I believe they are essential and should be explicitly reported.
2) The presentation of the foot model needs to be more detailed since it also comnstitute a particular methodological choice, particularly in comparison to standard modeling approaches. Specifically, it would be valuable to explore how the proposed model differs from the standard modeling approaches suggested by Dumas, Raphaël, Laurence Cheze, and J-P. Verriest in their paper titled “Adjustments to McConville et al. and Young et al. body segment inertial parameters” (Journal of Biomechanics, 40.3, 2007, pp. 543-553). 3) Equation (17) depicts the inverse dynamics equation for obtaining joint torques. However, it is essential to explicitly state that during the inverse dynamics phase, the problem of joint force estimation arises. Additionally, from an algorithmic perspective, the authors should provide details about the steps and equations used to compute the ground reaction forces. 4)The inertia matrices for each segment play a crucial role in the inverse dynamics problem. Unfortunately, the authors did not provide sufficient information regarding this aspect. It is essential to specify how they constructed the inertia tensors and with respect to which reference frame they are described. I recommend addressing these aspects following the guidelines outlined in the previously mentioned paper. 5) I recommend that the authors present two types of plots. First, the one already reported, which aids readers in understanding how the model behaves across different subjects. Second, they should depict the intrasubject behavior over repeated gait cycles (not just a single cycle).Minor Concerns
1) In the introduction, I recommend that the authors mention that, under machine learning approaches for estimating kinetic data, one can utilize EMG or a combination of EMG and kinematic data. This approach has been recently demonstrated in the work by Mobarak, Rami, et al. titled “A Minimal and Multi-Source Recording Setup for Ankle Joint Kinematics Estimation During Walking using only Proximal Information from Lower Limb” (published in IEEE Transactions on Neural Systems and Rehabilitation Engineering, 2024).
2) I suggest to add a qq-plot of the estimeted residual to show if they are normally distributed.
3) When presenting plots of moments and forces, please adhere to the standards proposed in the paper by Derrick, Timothy R., et al. titled “ISB recommendations on the reporting of intersegmental forces and moments during human motion analysis” (published in the Journal of Biomechanics, volume 99, 2020, page 109533).
Reviewer 3 Report
Comments and Suggestions for Authors
Dear Authors,
for possible acceptance of the article, it is necessary to move a number of information (including figures) to the results section
Reviewer 4 Report
Comments and Suggestions for Authors
This article proposes a method based on optical motion capture, which shortens the time for calculating ground reaction forces, ground reaction moments, and joint moments. However, the theoretical aspect does not highlight optical motion capture and needs to be fully supplemented to convince readers.
Comments on the Quality of English Languageno
Round 2
Reviewer 1 Report
Comments and Suggestions for Authors
The authors have addressed all concerns. I recommend it for publication.
Author Response
Thank you for reviewing our study. We have made minor revisions to the manuscript.
Reviewer 2 Report
Comments and Suggestions for Authors
Authors have addressd my main concerns. Good work.
Author Response
Thank you for reviewing our study. We have revised the figures and tables based on ISB recommendations. However, we have kept the normalized method unchanged in the paper because we used the same method as in previous studies of GRF estimation for comparison purposes.
Regarding the qq-plot, we decided not to include it in this paper. Our study conducted a correlation analysis between the estimated and measured results rather than a regression analysis. Therefore, we determined that a qq-plot indicating the normality of the residuals is not necessary for this paper. If you have any additional comments or feedback, we would appreciate it.
Reviewer 3 Report
Comments and Suggestions for Authors
Dear Authors,
you don´t include my recommendation in the revised version of manuscript. From this reason I recomment reject your manuscript.
Author Response
I have already revised the manuscript based on your comment. If you have any concerns, please provide the details.